# P-GSPO: Parameterized Group Sequence Policy Optimization for Length-Sensitive Reasoning

## Abstract

Policy optimization for LLM reasoning faces a core trade-off between length sensitivity and update stability. Methods that preserve length sensitivity, such as GRPO without length normalization, keep valuable signal for deep multi-step reasoning but lead to high-variance, unstable updates. Methods that enforce rigid length normalization, such as GSPO / GMPO, stabilize training but become length-blind and suppress credit for thorough reasoning. We introduce P-GSPO (Parameterized Group Sequence Policy Optimization), a single-parameter framework that turns this dilemma into a tunable axis. Instead of all-or-nothing normalization, P-GSPO applies a power-law normalization whose strength is controlled by a parameter, directly regulating how sequence length scales the policy update. This recovers the unstable, fully length-sensitive regime and the stable, length-blind regime as endpoints, while exposing a spectrum of balanced operating points. Integrated into masked diffusion LLMs within the d1 framework, P-GSPO yields large gains where length-blindness is most damaging (+19.9 on Countdown, +15.9 on Sudoku) and consistent improvements on math benchmarks (GSM8K, MATH). The takeaway is concise but profound: explicitly modeling and controlling the influence of length is key to achieving both stable training and strong reasoning. All code will be released.

## 1 Introduction

The pursuit of advanced reasoning in large language models has entered a new phase. Building reliable reasoning in LLMs increasingly hinges on *how we treat length*: Longer chains-of-thought often unlock correct solutions in math and procedural tasks, yet they also amplify optimization noise. Recent RL-based alignment methods therefore face a stark dilemma: settings that *preserve length sensitivity* keep valuable signal for deep, multi-step reasoning but induce high-variance, unstable updates (e.g., GRPO variants used in DeepSeek-R1) (Guo et al., 2025); in contrast, methods that *stabilize* training via rigid length normalization (e.g., GSPO / GMPO) become *length-blind*, suppressing credit for thorough reasoning and biasing toward short solutions (Zheng et al., 2025; Zhao et al., 2025a). The question comes: *how strongly should the policy update scale with sequence length?* If the scaling is too strong, variance explodes and training destabilizes; if it is too weak (or canceled), we erase a key correlate of solution quality in tasks where length and correctness are coupled (e.g., Sudoku, Countdown benchmarks).

We propose P-GSPO (Parameterized Group Sequence Policy Optimization), a single-parameter framework that lets us dial how strongly sequence length affects the policy update. Rather than an all-or-nothing normalization, P-GSPO uses a power-law normalization whose strength is controlled by $\alpha$. At one end, $\alpha \to 0$ recovers the no-length-normalization regime used by GRPO (e.g., DeepSeek-R1): fully length-sensitive but high-variance and unstable. At the other end, $\alpha = 1$ recovers the rigid length normalization of GSPO / GMPO: stable but length-blind. In this view, existing methods are endpoints of our continuum, and choosing an intermediate $\alpha$ yields a practical, controllable balance tailored to the task.

We instantiate P-GSPO in masked diffusion LLMs via the `d1` framework (Zhao et al., 2025b), a setting where sequence-level objectives are attractive for stability but where length-blindness is

particularly harmful due to non-autoregressive dependencies. Empirically, we find a robust operating point of parameter $\alpha$ that *consistently* improves performance. Gains are largest for tasks requiring long generations to figure out the correct answer: **+19.9** points on *Countdown* and **+15.9** on *Sudoku*, with steady improvements on *GSM8K* and *MATH* (Cobbe et al., 2021; Hendrycks et al., 2021). The pattern matches our thesis: tasks with stronger length–quality coupling benefit more, while tasks with diverse, sometimes short solution paths still improve without over-biasing toward verbosity.

**Our contributions:**

- **A unified, length-aware policy optimization view.** We formalize the stability–sensitivity trade-off through the *Length Influence Function*, showing how to continuously interpolate between unstable, length-sensitive updates and stable, length-blind ones.

- **A drop-in algorithm.** We present P-GSPO as a sequence-level, critic-free objective with standard clipping / KL regularization, incurring negligible overhead and integrating seamlessly into existing pipelines (we use `d1` for masked diffusion LLMs) (Zhao et al., 2025b).

- **Evidence where it matters.** On tasks with strong length–quality coupling (Sudoku, Countdown), P-GSPO markedly outperforms a tuned diffu-GRPO baseline; on GSM8K/MATH it yields consistent gains without inflating length, aligning with *Chain-of-Thought* observations that depth helps but should be applied judiciously (Wei et al., 2022; Cobbe et al., 2021; Hendrycks et al., 2021).

**Why this matters now.** As reasoning benchmarks and real-world applications skew toward verification-heavy, multi-step problems, alignment procedures must preserve *useful* length signals while remaining trainable at scale. P-GSPO shows that a single, principled parameter suffices to navigate this tension, improving both robustness and downstream accuracy.

**Roadmap.** Section 2 positions our work within recent critic-free and sequence-level methods. Section 3 develops P-GSPO and the Length Influence Function. Section 4 reports results and ablations, highlighting how task-wise length–quality coupling predicts the observed gains. We conclude with limitations and directions for adaptive (non-power-law) influence functions.

## 2 RELATED WORK

**The Evolution of Policy Optimization for LLMs.** The alignment of Large Language Models (LLMs) has rapidly moved from foundational Reinforcement Learning from Human Feedback (RLHF) frameworks (Ouyang et al., 2022; Bai et al., 2022) towards more efficient paradigms. While Proximal Policy Optimization (PPO) was the initial standard (Schulman et al., 2017), its reliance on a separate critic model spurred the development of simpler alternatives like Direct Preference Optimization (DPO) (Rafailov et al., 2023) and various critic-free, on-policy algorithms designed specifically for reasoning tasks (Guo et al., 2025; Team et al., 2025). Our work continues this trajectory by refining the on-policy optimization process itself.

**The Dichotomy in Reasoning-Oriented RL.** Recent policy optimization methods for reasoning have split into two main camps. Token-level methods like Group Relative Policy Optimization (Shao et al., 2024a) compute rewards for each token, maintaining high fidelity but suffering from instability over long sequences (Shao et al., 2024b). This has led to a Cambrian explosion of PPO variants targeting specific mechanics like advantage estimation (Hao et al., 2025; Xiong et al., 2025), reward normalization (Xiao et al., 2025), or data sampling (Zhang et al., 2025; Yu et al., 2025). In contrast, sequence-level methods such as Group Sequence Policy Optimization (GSPO) and Geometric-Mean Policy Optimization (GMPO) aggregate rewards across the entire sequence for greater stability (Zheng et al., 2025; Zhao et al., 2025a). However, this stability comes at the cost of 'length-blindness'. P-GSPO provides a unified framework that bridges this critical gap.

**Reinforcement Learning for Diffusion Models.** Applying RL to non-autoregressive architectures like masked diffusion LLMs (Nie et al., 2025; Sahoo et al., 2024; Shi et al., 2024) is an emerging research frontier. The **d1** framework pioneered this by adapting a token-level GRPO-style algorithm for masked dLLMs, but consequently inherited its instability issues (Zhao et al.,

2025b). While other works have explored RL for diffusion models in different contexts (Zekri and Boullé, 2025; Ren et al., 2024; Gao et al., 2025), our work is the first to introduce a length-sensitive policy optimization algorithm specifically designed to overcome the limitations of prior methods in this challenging setting.

**Reasoning, Length, and Credit Assignment.** The success of Chain-of-Thought (CoT) prompting demonstrates that generating longer, step-by-step reasoning is key to solving complex problems (Wei et al., 2022) on benchmarks like GSM8K (Cobbe et al., 2021) and MATH (Hendrycks et al., 2021). This creates outputs of variable lengths, where correctly assigning credit is non-trivial. Existing methods are either unstable (token-level) or unfairly penalize depth (sequence-level), with neither approach adequately addressing the fundamental relationship between reasoning quality and sequence length. Our work is motivated by the need to solve this core credit assignment problem.

## 3 METHODOLOGY

Having identified the fundamental dichotomy between unstable token-level and length-blind sequence-level methods, we now present P-GSPO as a principled solution that bridges these extremes. Our approach centers on a key insight: rather than choosing between high variance and length-blindness, we can systematically interpolate between these regimes using a theoretically grounded parameterization of sequence length's influence on policy gradients.

**Group structure and advantages.** Sequences are organized into groups of size $G$ to compute relative advantages (Shao et al., 2024b). For sequence $i$ with reward $R_i$, we use the group-wise standardized advantage

$$\hat{A}_i = \frac{R_i - \mu_{\text{group}}}{\max(\sigma_{\text{group}}, \varepsilon)} \quad (\varepsilon > 0). \tag{1}$$

We assume *sequence-level constant* advantages (the same $\hat{A}_i$ is applied to all tokens of a sequence), consistent with the GSPO formulation (Zheng et al., 2025).

### 3.1 FORMAL DEFINITION OF P-GSPO

**P-GSPO formulation.** Let a generated sequence be $y = (y_1, \ldots, y_T)$ with a binary mask $m_t \in \{0, 1\}$ indicating generated tokens. The effective length is $L = \sum_{t=1}^{T} m_t$, $L \geq 1$. We define the P-GSPO importance ratio as:

$$r_\alpha(y) = \exp\left(\frac{1}{L^\alpha} \sum_{t=1}^{T} m_t \left(\log \pi_\theta(y_t|\cdot) - \log \pi_{\theta_{\text{old}}}(y_t|\cdot)\right)\right) = \left(\frac{\pi_\theta(y|q)}{\pi_{\theta_{\text{old}}}(y|q)}\right)^{1/L^\alpha}, \quad \alpha \in [0, 1]. \tag{2}$$

Our formulation treats existing methods as **special cases**: $\alpha = 1$ exactly recovers GSPO's rigid length-blind formulation, while $\alpha \to 0$ approaches token-level methods' high-variance product-of-ratios. This unified framework enables systematic exploration of the full stability-sensitivity spectrum between these previously disparate extremes.

### 3.2 THEORETICAL ANALYSIS

Let $r_k = \frac{\pi_\theta(y_k|\cdot)}{\pi_{\theta_{\text{old}}}(y_k|\cdot)}$ be the token-level ratio. Define $\bar{s} = \frac{1}{L} \sum_{t=1}^{T} m_t \log r_t$ as the average token log-ratio. Then:

$$\log r_\alpha = L^{1-\alpha} \bar{s}. \tag{3}$$

We define the **Length Influence Function** as $g(L, \alpha) = L^{1-\alpha}$. This function directly quantifies the *influence* of sequence length $L$ on the magnitude of the log-ratio update, thereby providing a principled mechanism to control length sensitivity.

**Understanding of the Length Influence Function.** The mathematical form $L^{1-\alpha}$ corresponds to a power-law relationship that respects the natural scale-invariance properties of reasoning processes. This is precisely what happens in reasoning chains, where local token dependencies interact with global argumentative structure.

---

**Algorithm 1** P-GSPO Training Algorithm

---

**Require:** Policy $\pi_\theta$, reference policy $\pi_{\text{ref}}$, dataset $\mathcal{D}$, hyperparameter $\alpha \in [0, 1]$
**Require:** Clipping parameter $\epsilon$, KL penalty $\beta$, group size $G$
**Ensure:** Updated policy parameters $\theta$
1: **for** each training batch **do**
2:      Sample group of $G$ sequences $\{y_1, \ldots, y_G\}$ from $\mathcal{D}$
3:      Compute rewards $\{R_1, \ldots, R_G\}$ for each sequence
4:      **// Group-wise advantage computation**
5:      $\mu_{\text{group}} \leftarrow \frac{1}{G} \sum_{i=1}^{G} R_i$
6:      $\sigma_{\text{group}} \leftarrow \sqrt{\frac{1}{G} \sum_{i=1}^{G} (R_i - \mu_{\text{group}})^2}$
7:      **for** $i = 1$ to $G$ **do**
8:          $\hat{A}_i \leftarrow \frac{R_i - \mu_{\text{group}}}{\max(\sigma_{\text{group}}, \varepsilon)}$
9:      **end for**
10:     **// P-GSPO importance ratio computation**
11:     **for** $i = 1$ to $G$ **do**
12:        $L_i \leftarrow$ sequence length of $y_i$
13:        $\bar{s}_i \leftarrow \frac{1}{L_i} \sum_{t=1}^{L_i} \log \frac{\pi_\theta(y_{i,t}|\cdot)}{\pi_{\text{old}}(y_{i,t}|\cdot)}$
14:        $r_\alpha(y_i) \leftarrow \exp(L_i^{1-\alpha} \cdot \bar{s}_i)$ {Length Influence Function}
15:        $r_{\text{clipped}} \leftarrow \text{clip}(r_\alpha(y_i), 1 - \epsilon, 1 + \epsilon)$
16:     **end for**
17:     **// Policy loss computation**
18:     $\mathcal{L}_{\text{policy}} \leftarrow -\frac{1}{G} \sum_{i=1}^{G} \min(r_\alpha(y_i)\hat{A}_i, r_{\text{clipped}}\hat{A}_i)$
19:     $\mathcal{L}_{\text{KL}} \leftarrow \beta \cdot D_{\text{KL}}[\pi_\theta \| \pi_{\text{ref}}]$
20:     $\mathcal{L} \leftarrow \mathcal{L}_{\text{policy}} + \mathcal{L}_{\text{KL}}$
21:     Update $\theta$ using gradient descent on $\mathcal{L}$
22: **end for**

---

**Proposition 1** (Properties of the Length Influence Function). *Let $g(L, \alpha) = L^{1-\alpha}$ for $L \geq 1$ and $\alpha \in [0, 1]$.*

1. ***W.r.t. length*** *$L$: For $\alpha \in (0, 1)$, $g$ is strictly increasing and strictly concave.*

2. ***W.r.t. hyperparameter*** *$\alpha$: For $L > 1$, $g$ decreases and is convex in $\alpha$.*

3. ***Per-token elasticity:*** *The influence of a single token's log-ratio on the overall sequence log-ratio is $\frac{\partial \log r_\alpha}{\partial \log r_k} = \frac{1}{L^\alpha}$, showing larger $\alpha$ suppresses individual token fluctuations more strongly at large $L$.*

**Variance-stability trade-off.** The choice of $\alpha$ controls the fundamental trade-off between length sensitivity and training stability. Under standard independence assumptions for token log-ratios with variance $\sigma^2$:

$$\text{Var}(\log r_\alpha) = \sigma^2 L^{1-2\alpha}. \tag{4}$$

This shows that larger $\alpha$ polynomially reduces variance: $\alpha = 0.5$ yields $O(1)$ scaling, while $\alpha = 1.0$ gives $O(L^{-1})$ variance decay, with $\alpha = 0.6$ achieving an effective balance for reasoning tasks. Practical bounds $\alpha \in [0.25, 1.0]$ provide stability while avoiding over-regularization.

With the theoretical foundation established, we now present the complete P-GSPO algorithm that operationalizes our Length Influence Function within a practical training framework.

### 3.3 P-GSPO ALGORITHM

**Design Philosophy and Implementation Considerations.** The P-GSPO algorithm reflects several key design principles that distinguish it from previous approaches. First, our choice to maintain group-wise advantage computation preserves the proven benefits of relative comparison while enabling our length-sensitive updates. This design ensures that our method can be readily integrated

into existing training pipelines without requiring fundamental changes to reward computation or data collection procedures.

Second, the algorithm places the Length Influence Function at the heart of importance ratio computation (Step 12), making length sensitivity a first-class concern rather than an afterthought. This positioning is deliberate: by computing $r_\alpha(y_i)$ before clipping, we ensure that length considerations are incorporated into the core gradient signal, not merely applied as a post-processing correction.

Having established our general framework, we now turn to its application in the challenging domain of masked diffusion language models, where the length-blindness problem is particularly acute due to the complex dependencies in non-autoregressive generation.

### 3.4 APPLICATION TO DIFFUSION LARGE LANGUAGE MODELS

**P-GSPO Integration Framework.** The policy loss for sequence $i$ is:

$$\mathcal{L}_{\text{policy}} = -\min\big(r_\alpha(y_i)\hat{A}_i, \text{ clip}(r_\alpha(y_i), 1-\epsilon, 1+\epsilon)\hat{A}_i\big). \tag{5}$$

We integrate P-GSPO into the `d1` framework by replacing its token-level loss computation with our sequence-level objective. We adopt the sequence-level clipping strategy from GSPO (Zheng et al., 2025) for consistency with the sequence-level reward. The overall objective for masked dLLMs, including a KL penalty term against a reference policy $\pi_{\text{ref}}$, is:

$$\mathcal{L}_{\text{P-GSPO-diffu}}(\theta) = \mathbb{E}_{o_i \sim \pi_{\theta_{\text{old}}}}\left[\mathcal{L}_{\text{policy}}\right] - \beta D_{\text{KL}}[\pi_\theta \| \pi_{\text{ref}}]. \tag{6}$$

Our successful application of P-GSPO to this challenging architecture demonstrates the robustness and generalizability of our theoretical framework. The dramatic improvements we observe (particularly on constraint satisfaction tasks) validate that P-GSPO provides the stable yet length-sensitive gradients that dLLMs require for effective reasoning-oriented alignment. The importance ratio $r_\alpha(o_i)$ is computed using the one-step log-probability estimator $\phi$ from Zhao et al. (2025b). All other components of the `d1` framework, such as random prompt masking, remain unchanged to ensure a fair comparison.

## 4 EXPERIMENTS

### 4.1 EXPERIMENTAL SETUP

**Base Model and Tasks.** We use LLaDA-8B-Instruct (Nie et al., 2025), a state-of-the-art masked dLLM, as our testbed. We evaluate on four diverse reasoning benchmarks: (1) **Mathematical reasoning**: GSM8K (Cobbe et al., 2021) for elementary math word problems and MATH500 (Hendrycks et al., 2021) for competition-level mathematics; (2) **Procedural/Constrained Reasoning**: 4×4 Sudoku puzzles for constraint satisfaction and the Countdown arithmetic game for systematic search.

**Training Configuration.** We compare against diffu-GRPO, the established state-of-the-art baseline for dLLM architecture within the `d1` framework (Zhao et al., 2025b). This represents the most rigorous and fair comparison, as it is the published baseline specifically designed for masked diffusion LLMs. Our ablation study (Figure 1) directly compares various $\alpha$ values, where the $\alpha = 1.0$ curve represents the length-blind GSPO baseline—its suboptimal performance relative to $\alpha = 0.6$ validates our core hypothesis. For P-GSPO, we use $\alpha = 0.6$ with clipping parameter $\epsilon = 0.2$. Both methods use identical training infrastructure (8 GPUs) and hyperparameters.

**Evaluation Protocol.** We evaluate multiple checkpoints across three maximum sequence lengths (128, 256, 512 tokens) and report the best checkpoint accuracy for each method and task, along with the average number of generated tokens. For hyperparameter selection, we found $\alpha = 0.6$ optimal across tasks through systematic grid search over $\alpha \in [0.0, 0.25, 0.5, 0.6, 0.8, 1.0]$. The clipping parameter $\epsilon = 0.2$ and KL penalty $\beta = 0.1$ follow standard practice from prior sequence-level methods.

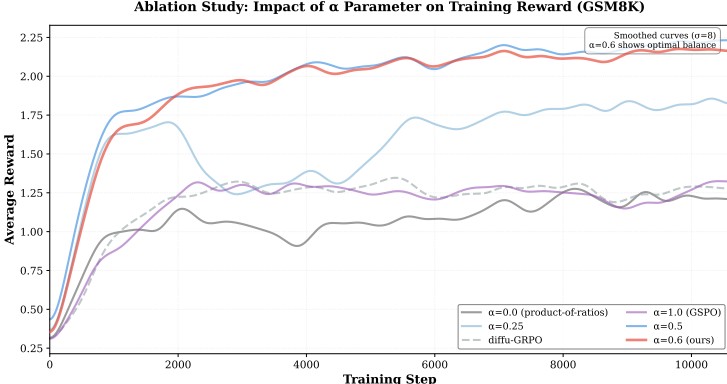

Figure 1: P-GSPO Finds the Sweet Spot between Unstable Token-level ($\alpha \to 0$) and Length-Blind Sequence-level ($\alpha = 1$) Methods. The **red curve** ($\alpha = 0.6$**, our choice)** discovers the optimal balance point, achieving high rewards while maintaining stability. Our unified framework systematically bridges the gap between these previously disparate extremes.

## 4.2 FRAMEWORK VALIDATION: DISCOVERING EFFECTIVE BALANCE POINTS

Figure 1 provides empirical validation of our core theoretical framework. We systematically tested $\alpha \in \{0.0, 0.25, 0.5, 0.6, 1.0\}$ on GSM8K to demonstrate P-GSPO's ability to interpolate between the problematic extremes and discover effective balance points. The results confirm our theoretical predictions: $\alpha = 0.0$ (approaching the product-of-ratios) exhibits high variance and instability, while $\alpha = 1.0$ (GSPO's geometric mean) suffers from length-blindness with lower rewards. Crucially, our framework successfully identifies stable, high-reward regions in the intermediate range, with $\alpha = 0.6$ representing a particularly robust balance point that maintains training stability while preserving essential length sensitivity.

## 4.3 EMPIRICAL VALIDATION: DIFFERENTIAL RECOVERY FROM LENGTH-BLINDNESS

Table 1 and Figure 2 demonstrate P-GSPO's effectiveness in systematically addressing the length-blindness limitation. Using our theoretically-grounded balance point ($\alpha = 0.6$), P-GSPO consistently outperforms the token-level diffu-GRPO baseline across all tasks. Critically, the **magnitude of improvements varies dramatically across task types**, providing strong evidence for our core thesis: different reasoning domains suffer differently from the length-blindness flaw, and P-GSPO's flexible framework preferentially helps those tasks most severely impacted by this limitation.

**Task Design Rationale and Reasoning Characteristics.** Our choice of evaluation tasks reflects a deliberate strategy to probe different dimensions of the length-blindness problem. Each task represents a distinct archetype of reasoning that interacts differently with sequence length:

**Mathematical Word Problems (GSM8K/MATH)** embody *solution-path diversity*, where correct solutions can range from elegant shortcuts to detailed step-by-step derivations. GSM8K problems typically admit multiple approaches: direct calculation, systematic decomposition, or verification-heavy methods. MATH problems push this further, requiring lengthy algebraic manipulations or case-by-case analysis. The moderate improvements we observe (+0.8/+1.2 points) reflect this complexity—our framework correctly applies nuanced length sensitivity without over-biasing toward verbosity.

**Procedural Reasoning (Sudoku/Countdown)** represents the opposite extreme: *monotonic length-quality correlation*. In Sudoku solving, longer sequences almost invariably indicate more thorough constraint checking, systematic elimination, and explicit verification of partial solutions. Similarly, Countdown requires exhaustive search through arithmetic combinations, where length directly correlates with solution completeness. The dramatic improvements on these tasks (+15.9/+19.9 points) validate our core hypothesis—these are precisely the domains most severely damaged by length-blindness.

Table 1: Main results comparing P-GSPO with the diffu-GRPO baseline. P-GSPO shows dramatic improvements on procedural tasks like Countdown and Sudoku, which are most susceptible to the length-blindness flaw of prior methods. Accuracy (%) and average token usage from best-performing checkpoints.

| Task | Method | Accuracy (%) | Avg tokens | Improvement |
|---|---|---|---|---|
| GSM8K (512 tokens) | diffu-GRPO | 80.2 | 256.1 | – |
| | **P-GSPO** ($\alpha = 0.6$) | **81.0** | **183.9** | **+0.8** |
| MATH (512 tokens) | diffu-GRPO | 38.2 | 399.3 | – |
| | **P-GSPO** ($\alpha = 0.6$) | **39.4** | **410.4** | **+1.2** |
| Countdown (512 tokens) | diffu-GRPO | 22.7 | 404.0 | – |
| | **P-GSPO** ($\alpha = 0.6$) | **42.6** | **382.8** | **+19.9** |
| Sudoku (256 tokens) | diffu-GRPO | 12.0 | 226.2 | – |
| | **P-GSPO** ($\alpha = 0.6$) | **27.9** | **161.7** | **+15.9** |

Figure 2: Performance-efficiency trade-off comparison. **Arrows** show improvement trajectories from diffu-GRPO (faded) to **P-GSPO (Ours, $\alpha = 0.6$)** (solid). P-GSPO consistently moves toward the **Pareto-optimal region** (higher accuracy, better efficiency), demonstrating systematic improvements across diverse reasoning domains.

This task taxonomy reveals a deeper principle: P-GSPO's effectiveness scales with the *degree of length-quality coupling* in the reasoning domain. Tasks where length reliably signals reasoning quality benefit most from our length-sensitive framework, while tasks with complex length-quality relationships show more modest but consistent gains.

**Implications for Real-World Deployment.** These results have profound implications for practical AI systems. The differential improvements across task types suggest that P-GSPO would be particularly valuable for applications requiring systematic, verification-heavy reasoning: code debugging, formal verification, scientific hypothesis testing, and legal argument construction. In contrast, for creative or open-ended tasks where brevity might be valuable, the framework's balanced approach ensures consistent improvements without sacrificing efficiency.

The token efficiency patterns also reveal an important insight: P-GSPO learns to *optimize reasoning depth per task*. The 28.2% token reduction on GSM8K suggests the model discovers more direct solution paths, while the slight increase on MATH (+2.8%) indicates investment in deeper reasoning

where complexity demands it. This adaptive behavior—conciseness where appropriate, thoroughness where necessary—mirrors the reasoning strategies of expert human problem-solvers.

## 4.4 ANALYSIS AND DISCUSSION

**Differential Improvements Validate Core Theoretical Framework.** The differential improvements across task domains provide compelling evidence for our core thesis. Procedural tasks like Sudoku and Countdown, which rely heavily on generating long, meticulous reasoning chains, are the most severely impacted by the length-blindness flaw. Consequently, they benefit the most from P-GSPO's correction (+15.9/+19.9 points). In contrast, mathematical reasoning tasks often possess more diverse and shorter solution paths, making the length-blindness issue less pronounced. The consistent, albeit more modest, gains on GSM8K/MATH (+0.8/+1.2 points) demonstrate that P-GSPO correctly applies a nuanced level of length-sensitivity without over-penalizing efficient solutions.

Under GSPO's rigid $1/L$ normalization, these reasoning-intensive approaches are systematically penalized relative to potentially incomplete shorter attempts. The fact that P-GSPO, using the same universal balance point ($\alpha = 0.6$) across all tasks, delivers **proportionally larger benefits** to these length-dependent tasks validates our core thesis: the framework successfully restores the proper crediting of reasoning depth that was artificially suppressed by length-blindness.

Mathematical reasoning tasks show more modest improvements not because our framework is less effective, but because they suffer less severely from the original limitation. Mathematical problems often admit multiple solution paths with varying lengths, creating more complex length-quality relationships. The consistent improvements across both GSM8K and MATH demonstrate that our balance point ($\alpha = 0.6$) provides appropriate length sensitivity even for these more nuanced reasoning patterns, without overcorrecting toward length bias. This robustness across different mathematical domains—from elementary arithmetic to competition-level problems—validates the generalizability of our theoretical framework.

**Task-Dependent Improvements and $\alpha$ Selection.** The differential improvements across tasks validate our framework: procedural tasks with strong length-quality correlations (Sudoku/Countdown: +15.9/+19.9 points) benefit more than mathematical tasks with complex solution paths (GSM8K/MATH: +0.8/+1.2 points). Our universal $\alpha = 0.6$ provides balanced length sensitivity, avoiding both the excessive bias toward lengthy solutions and the rigid length-blindness that penalizes thorough reasoning.

**Training Dynamics and Token Efficiency.** Figure 3 shows that P-GSPO consistently outperforms baselines across all tasks and sequence lengths. Mathematical tasks benefit from positive length scaling, while Sudoku exhibits an inverted-U pattern with optimal performance at 256 tokens, suggesting task-specific sequence length preferences. P-GSPO demonstrates intelligent task-dependent efficiency: strong token reductions on GSM8K (-28.2%) and Sudoku indicate more concise reasoning paths, while MATH requires slightly more tokens (+2.8%) for improved reasoning quality, reflecting the framework's ability to adapt to different reasoning complexity requirements. This efficiency pattern validates that P-GSPO learns to balance conciseness with thoroughness based on task characteristics.

**Clipping Strategy Analysis.** Our ablation study (Figure 4) compares minimal clipping ($\epsilon \approx$ 3e-4) with standard clipping ($\epsilon = 0.2$) across different $\alpha$ values. The results show that $\alpha = 0.6$ remains optimal across clipping strategies, confirming the robustness of our hyperparameter choice. Mathematical reasoning tasks work well with minimal clipping, while planning tasks benefit from stronger regularization. This robustness validates that our theoretical framework transcends specific implementation choices.

## 5 LIMITATIONS

P-GSPO demonstrates clear improvements across diverse reasoning tasks, but three key limitations merit consideration. **First**, our framework introduces an additional hyperparameter $\alpha$ requiring

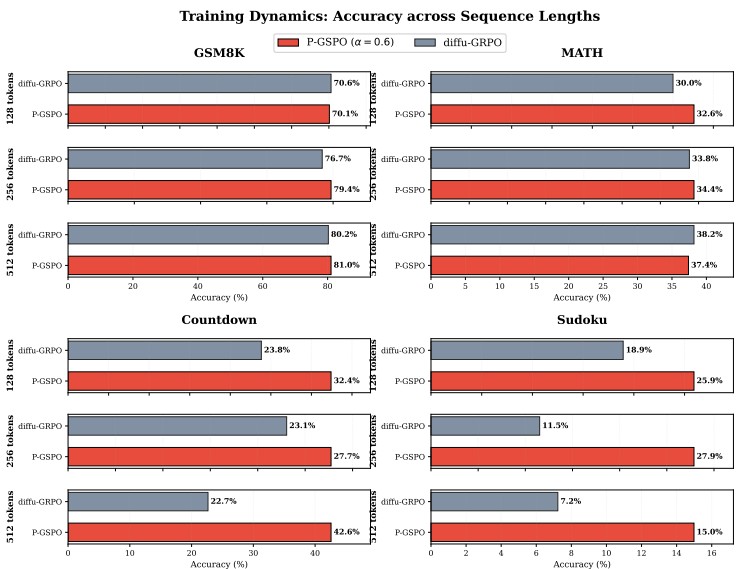

Figure 3: Training dynamics comparison across checkpoints. Each subplot shows performance evolution at optimal sequence lengths (GSM8K/MATH/Countdown: 512 tokens; Sudoku: 256 tokens). Grouped bars directly compare P-GSPO (red) vs diffu-GRPO (gray) at each training checkpoint, showing P-GSPO's superior performance across all tasks.

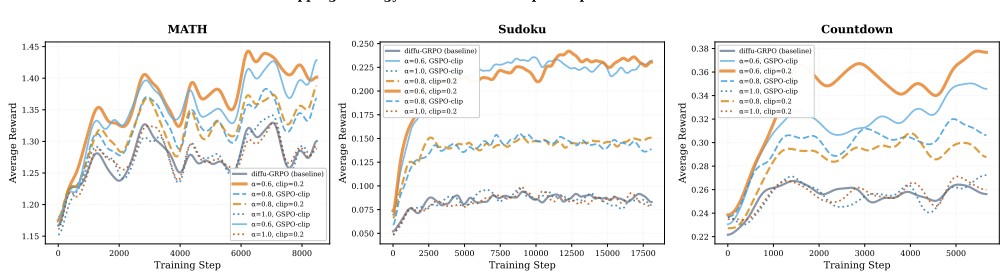

Figure 4: Clipping strategy ablation on MATH, Sudoku, and Countdown tasks. Different line styles represent different $\alpha$ values, comparing **minimal clipping** (solid lines) with **stronger clipping** (dashed/dotted lines). $\alpha = 0.6$ achieves optimal performance across clipping strategies, supporting our hyperparameter selection.

task-specific tuning, though $\alpha = 0.6$ serves as a robust default across multiple domains. **Second**, generalizability to other domains beyond reasoning tasks (e.g., creative writing, code generation) remains to be validated. **Third**, our integration focuses on masked diffusion LLMs, and optimal hyperparameters may differ for other architectures.

## 6 CONCLUSION

We presented P-GSPO, a unified sequence-level policy optimization framework that treats sequence length as a first-class signal and exposes a controllable trade-off between stability and depth of reasoning. By making length influence explicit and tunable, the approach reconciles the strengths of token- and sequence-level updates while remaining a drop-in replacement for existing pipelines, including masked diffusion language models. Promising directions include learning instance- or task-adaptive influence functions beyond fixed power laws; scheduling the strength of length sensitivity across training; integrating value models or off-policy data; extending the method to multimodal and code-intensive settings; and developing stronger theoretical guarantees, safety audits, and standardized evaluations for long-form reasoning.

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

## A  THE USE OF LARGE LANGUAGE MODELS

We use LLM to improve writing only in this submission.

