# OpenReview forum: "P-GSPO: Parameterized Group Sequence Policy Optimization for Length-Sensitive Reasoning"
_ICLR.cc/2026/Conference — ICLR 2026 Conference Withdrawn Submission_

### Official Review · Reviewer_xMBL · 2025-10-29

**Soundness:** 3
**Presentation:** 3
**Contribution:** 3
**Rating:** 6
**Confidence:** 2

**Summary:**

This paper proposes P-GSPO, a parameterized variant of GSPO to balance length sensitivity and training stability in reinforcement learning for reasoning tasks. The idea is to introduce a tunable exponent $\alpha$ that interpolates between fully length-sensitive and length-blind updates. The paper shows promising results on masked diffusion LLMs for both synthetic and math reasoning benchmarks such as GSM8K and MATH.

**Strengths:**

- The paper tackles an important problem on accuracy and length tradeoff with a good explanation on length normalization.
- The paper is well-written with both synthetic and real-world dataset experiments on several reasoning benchmarks.

**Weaknesses:**

- The experiments are limited only on masked diffusion LLMs, with no results presented for decoder only LLMs.
- Claims of stable training are not supported by quantitative stability measures, making it hard to assess. I am also not sure if Figure 4 shows stable training or not.
- I am also a bit confused by the explanation between Line 318-323. I am under the impression that for Sudoku and countdown, the authors suggest the longer the response is, the better the performance is. However, Table 1 shows that for these tasks, shorter response actually significantly outperforms the baseline.

**Questions:**

1. Can authors show effectiveness of their approach on models like Llama or Qwen under LLM post-training framework? If the proposed method is generally applicable to length normalization, it should show improvement for both frameworks.
2. Can authors quantify it (e.g., regress correctness on length under different $\alpha$, or equal-length controls) to show gains aren’t just verbosity? That is, given the same sampling budget, the method still shows an advantage.

---

### Official Review · Reviewer_H9mR · 2025-10-31

**Soundness:** 3
**Presentation:** 3
**Contribution:** 3
**Rating:** 4
**Confidence:** 3

**Summary:**

The paper proposes P-GSPO, a parameterized extension of Group Sequence Policy Optimization that allows controlling how sequence length influences the policy update. The key idea is a single parameter alpha that interpolates between the unstable length-sensitive regime (GRPO) and the stable, length-blind regime (GSPO). The method is tested within masked diffusion LLMs using the d1 framework and evaluated on some reasoning benchmarks.

**Strengths:**

The motivation is clear: current methods either lose important length information or are unstable. The theory is solid, with a clear formulation of the length influence function and its trade-offs. The paper is easy to follow, and the algorithm is simple to use. Experiments cover several reasoning tasks and show improvements, especially on procedural tasks where length and quality are closely linked.

**Weaknesses:**

#### Major comments

My main concern is with the experimental section, which feels incomplete.


- Section 4.1 mentions that the best checkpoint among different sequence lengths is used, but from Table 1 and Figure 3 this seems true only for P-GSPO, not for diffu-GRPO. This makes the comparison a bit unclear.

- The diffu-GRPO results appear lower than those in Zhao et al. (2025b). Are these reproduced numbers or from the original paper? If reproduced, do they use identical hyperparameters?

- Table 1 and Figure 2 display nearly identical information (accuracy vs. tokens). One of them could be removed or replaced with something more informative, such as a deeper ablation.

- Since $\alpha$=0 and $\alpha$=1 correspond to GRPO and GSPO, it would be helpful to include those cases into table 1 or Figure 2 explicitly to verify that P-GSPO smoothly recovers both limits.

- The MATH numbers in Figure 3 do not seem to match those in Table 1. It’s not clear if they come from different checkpoints or sequence lengths.

- All experiments use diffusion-based LLMs. It would strengthen the paper to include at least one test on an autoregressive model to show that the benefit is algorithmic, not tied to the architecture.

---
#### Minor comments

- In Figure 3, some figure labels overlap and are hard to read.

- In Figure 3, subplots use different x-axis scales, which makes it difficult to compare across sequence lengths.

**Questions:**

- How sensitive is performance to $\alpha$ around 0.6?
- Equation 4 assumes token log-ratios have independent variance \$\sigma^2\$. Is this realistic for reasoning chains? How does correlation between tokens affect the variance-stability trade-off?
- The diffu-GRPO baseline comes from Zhao et al. (2025b). What was their reported performance? Are the baseline numbers reproductions or taken from the original paper? If reproduced, are they using the same hyperparameters?

---

### Official Review · Reviewer_oVxN · 2025-10-31

**Soundness:** 2
**Presentation:** 2
**Contribution:** 2
**Rating:** 2
**Confidence:** 4

**Summary:**

The paper proposes P-GSPO, a single-parameter sequence-level policy optimization method that interpolates between token-level product-of-ratios updates and sequence-level geometric-mean normalization. The core idea is to scale the importance ratio by a length influence function $g(L,\\alpha)=L^{1-\\alpha}$, yielding
$\\log r_\\alpha = L^{1-\\alpha} \\bar s,$
which makes the stability versus length sensitivity trade-off explicit; under independence assumptions, the variance scales as $\\mathrm{Var}(\\log r_\\alpha)=\\sigma^2 L^{1-2\\alpha}$, so larger $\\alpha$ reduces variance. The algorithm computes $r_\\alpha$ before clipping and uses GSPO-style sequence clipping with a KL penalty in the d1 masked diffusion LLM framework. Empirically, with $\\alpha=0.6$ the method improves over a diffu-GRPO baseline on GSM8K, MATH500, Countdown, and Sudoku, with large gains on procedural tasks; however, the reported diffu-GRPO baseline is substantially below the original d1 numbers on Countdown, which weakens the strength of the claimed gains.

**Strengths:**

* Clear formalization of the length influence function $g(L,\\alpha)$ with explicit variance scaling.
* Simple drop-in modification to the GSPO objective with clipping and KL.

**Weaknesses:**

* Countdown baseline appears under-trained relative to d1. P-GSPO reports diffu-GRPO 22.7 percent at 512, while d1 reports 37.1 percent for the same model and length. The main claimed gains on Countdown depend on this gap.
* Training hyperparameters and reward compositions for the reproduced diffu-GRPO baseline are not specified; only “identical infrastructure (8 GPUs)” is stated. Key d1 settings that affect outcomes are documented in detail but not confirmed here.
* No appendix with full configs or scripts is provided to verify reproducibility. The PDF ends at references.

**Questions:**

* Please reconcile the Countdown baseline: why is your diffu-GRPO at 512 tokens 22.7 percent while d1 reports 37.1 percent under the same model and evaluation length? Specify all training hyperparameters, reward components, steps, and checkpoint selection for this baseline.
* Did you match the d1 diffu-GRPO training settings, including LoRA rank and scaling, learning rate $5\\times10^{-6}$, $p_{\\text{mask}}=0.15$, and per-task step counts (GSM8K 7700, MATH500 6600, Countdown 5000, Sudoku 3800)? If not, please detail deviations.

---

### Official Review · Reviewer_YJwT · 2025-11-01

**Soundness:** 2
**Presentation:** 2
**Contribution:** 3
**Rating:** 4
**Confidence:** 4

**Summary:**

The paper proposes Parameterized GSPO (P-GSPO), a sequence-level policy-optimization objective for reasoning RL that introduces a single parameter $\\alpha$ to control length normalization of the log importance ratio. The authors analyze how $\\alpha$ affects a “length influence” term and the variance of the sequence ratio, and they instantiate P-GSPO in the d1 masked-diffusion LM setting with standard clipping/KL regularization. Empirically, they report more stable training and better outcomes on length-coupled reasoning tasks (and some math benchmarks) relative to GRPO/GSPO-style baselines.

**Strengths:**

- **Simple, clean approach.** The method exposes a single, interpretable parameter that lets practitioners choose the degree of length normalization of the log ratio.
- **Demonstrated effectiveness in diffusion LMs.** Experiments within the masked-diffusion LM setting show that the approach is workable and beneficial under that training regime.

**Weaknesses:**

- **Motivation is underdeveloped.** The paper briefly asserts that GSPO-like objectives “suppress credit for thorough reasoning” and bias toward short solutions, but does not concretely explain the mechanism or show evidence of the downstream consequences. The paper would be much stronger with the essential analysis (quoting other's analysis works as well) that motivates the proposed approach. Current one is weak.
- **Theory stops short of gradient-level implications.** The analysis focuses on the sequence-ratio and a length-influence function, but does not unpack how $\\alpha$ changes the gradient of the P-GSPO loss. Deriving or empirically probing the gradient scaling would help to clarify why/when P-GSPO improves optimization.
- **Scope limited to diffusion LMs without clear rationale.** The algorithm is presented generically but only evaluated (and seemingly targeted) for masked-diffusion LMs, with little discussion of why AR LMs would be unsuitable or how performance would translate. Providing results or at least a reasoned analysis for AR LMs (or explaining concrete obstacles) would be helpful.

**Questions:**

see weakness

---

### Note · Authors · 2025-11-14

I have read and agree with the venue's withdrawal policy on behalf of myself and my co-authors.